# *Campylobacter jejuni* in Different Canine Populations: Characteristics and Zoonotic Potential

**DOI:** 10.3390/microorganisms9112231

**Published:** 2021-10-26

**Authors:** Maria-Leonor Lemos, Alexandra Nunes, Massimo Ancora, Cesare Cammà, Paulo Martins da Costa, Mónica Oleastro

**Affiliations:** 1ICBAS—Institute of Biomedical Sciences Abel Salazar, University of Porto, 4050313 Porto, Portugal; up201305391@edu.icbas.up.pt (M.-L.L.); pmcosta@icbas.up.pt (P.M.d.C.); 2Infectious Diseases Department, National Institute of Health Dr. Ricardo Jorge, 1649016 Lisbon, Portugal; alexandra.nunes@insa.min-saude.pt; 3Faculty of Veterinary Medicine, Lusófona University, 1300477 Lisboa, Portugal; 4CBIOS—Universidade Lusófona’s Research Center for Biosciences & Health Technologies, Campo Grande 376, 1749024 Lisbon, Portugal; 5Istituto Zooprofilattico Sperimentale dell’Abruzzo e del Molise “G. Caporale”, 64100 Teramo, Italy; m.ancora@izs.it (M.A.); c.camma@izs.it (C.C.)

**Keywords:** *Campylobacter* spp., dogs, whole genome sequencing, One Health

## Abstract

With most epidemiological studies focused on poultry, dogs are often overlooked as a reservoir of *Campylobacter*, even though these animals maintain close daily contact with humans. The present study aimed to obtain a first insight into the presence and characteristics of *Campylobacter* spp. in different canine populations in Portugal, and to evaluate its zoonotic potential through genomic analysis. From a total of 125 rectal swabs collected from companion (*n* = 71) and hunting dogs (*n* = 54) living in two different settings, rural (*n* = 75) and urban (*n* = 50), 32 *Campylobacter* spp. isolates were obtained. Four different *Campylobacter* species were identified by Multiplex PCR and MALDI-TOF mass spectrometry, of which *Campylobacter jejuni* (*n* = 14, 44%) was overall the most frequently found species. Relevant resistance phenotypes were detected in *C. jejuni,* with 93% of the isolates being resistant to ciprofloxacin, 64% to tetracycline, and 57% to ampicillin, and three isolates being multi-drug-resistant. Comparison of the phenotypic and genotypic traits with human isolates from Portuguese patients revealed great similarity between both groups. Particularly relevant, the wgMLST analysis allowed the identification of isolates from human and dogs without any apparent epidemiological relationship, sharing high genetic proximity. Notwithstanding the limited sample size, considering the high genomic diversity of *C. jejuni,* the genetic overlap between human and dog strains observed in this study confirmed that the occurrence of this species in dogs is of public health concern, reinforcing the call for a One Health approach.

## 1. Introduction

Since 2005, campylobacteriosis is the most reported zoonotic disease in the European Union, mostly associated with *Campylobacter jejuni* and *Campylobacter coli* infections [1]. Being mainly a foodborne disease, it is frequently related to the consumption and handling of contaminated meat. Since ~80% of human infections are associated with poultry, epidemiological studies and prevention programs have largely focused on broilers [2,3]. Despite that, other sources of infection, such as wild animals, environmental water, and pets have been described and should also be considered [4].

In dogs, *Campylobacter upsaliensis* and *C. jejuni* are the predominantly found species, and despite being mainly considered as commensal organisms, with a prevalence of around 50% among subclinical animals, these are also described as pathogenic, especially in younger dogs, and are associated with clinical signs similar to those described in humans [5,6,7]. Owning a dog, particularly puppies, has been described as a risk factor for campylobacteriosis, and different studies around Europe have attributed 9–25% of human cases to pets [6,8,9,10,11,12]. The fact that neither dogs nor humans are considered primary sources or amplifying hosts of *Campylobacter* and that the close sharing of a living environment exposes them to common sources of infection, limits the assessment of the exact transmission route [8], which could lead to a bias in the aforementioned values. Despite that, there is evidence for dog–human transmission [13,14]. 

The ease with which bacteria and their resistance genes can spread among human and animal populations forces an extensive approach—based on the One Health concept—when facing this kind of zoonosis [15]. Therefore, comprehensive studies are essential, using appropriate typing methods, which allow an efficient assessment of the relationship between different strains and hosts. 

Since there was no previous study regarding *Campylobacter* spp. in dogs in Portugal, the present study aimed to obtain a first insight into the presence and characteristics of this microorganism in different canine populations, as well as to assess the zoonotic potential of the *Campylobacter jejuni* isolates through a genomic comparison with human isolates.

## 2. Materials and Methods

### 2.1. Sample Selection and Collection

To evaluate different canine populations, rectal swabs were collected from pet and hunting dogs, in two different settings, rural and urban. 

Seventy-one samples from household pets were collected from September to December 2020 from a Veterinary Hospital of the University of Porto (urban area) (*n* = 50) and a Veterinary Clinic located in a rural area in the north of Portugal (*n* = 21). Rectal swab specimens were obtained from dogs presented for consultation (*n* = 39), in pre- and post-surgery care (*n* = 16) and hospitalized (*n* = 16), on a convenience basis. The study design, including its ethical aspects, was reviewed and approved in advance with the report number P330/2019/ORBEA by the Ethical Committee of ICBAS (ORBEA) and by the Portuguese Animal Welfare Authority (DGAV). The owners were asked to sign a consent form and fill out a short questionnaire about the animal and its living environment, signs of gastrointestinal illnesses, and any recent antibiotic treatment. Among the 71 sampled dogs (male *n* = 32; female *n* = 39), 12 presented with a recent history of diarrhea, and five of them were in the hospital for that reason.

Samples of hunting dogs were collected from boarhounds and small game hounds from the same rural area as the Veterinary Clinic. Regarding the boarhounds, on November 23, 2020, two weeks after the opening of the boar hunting season, 17 (male *n* = 9; female *n* = 8) rectal swabs were collected from the same pack. Concerning this group, 12 dogs had already been hunting that season and five had not still. A month after the first sampling campaign, after all the animals had been hunting in “montarias” (typical Iberian boar-driven hunt), a second swab was collected from 15 of the initial 17 dogs sampled. Along with this boarhound pack, 22 (male *n* = 14; female *n* = 8) samples from small game hounds from the same area were obtained in November 2020 from three different packs (*n* = 8; *n* = 11; *n* = 3), totaling 54 specimens from hunting dogs. Regarding this group, none of the animals presented a history of gastrointestinal disease.

### 2.2. Culture and Isolation of Campylobacter spp.

Immediately after the rectal swab was performed, samples were placed into 10 mL of Bolton Broth (Bolton Broth with Bolton broth selective supplement, and 5% Horse blood Lysed, Oxoid, Madrid, Spain). Inoculated broths were incubated for 48 h, at 41.5 °C under microaerophilic conditions generated by CampyGen™ (Oxoid). After the enrichment, 120 μL of the suspension were placed in blood agar (BA) (Blood Agar: Trypto-casein soy agar, Biokar Diagnostics, supplemented with 5% Horse Lysed blood, Oxoid) through a 0.65 μm membrane filter (Nitrocellulose membrane filters, Whatman, Stuttgart, Germany) and subsequently incubated in the same conditions, for another 48-h period (ISO 10272:2006-1).

*Campylobacter*-like colonies (about five per sample) were tested for oxidase (Oxidase Test Stick, Liofilchem, Roseto degli Abruzzi, Italy) and growth under aerobic conditions, and morphology and motility evaluated through phase-contrast microscopy. DNA was extracted from the suspected colonies by the boiling method.

### 2.3. Isolate Identification

A PCR Multiplex *Campylobacter* was performed as described by Yamazaki-Matsune and colleagues (2007) for *Campylobacter* genus confirmation and species identification, targeting *C. jejuni*, *C. upsaliensis*, *C. coli*, *C. fetus*, *C. lari* and *C. hyointestinalis* [16]. PCR products were analyzed by gel electrophoresis through a 1.5% (weight/volume) agarose (Agarose Ultrapure grade, NZYtech, Lisbon, Portugal) in 1× TBE buffer, stained with 1% GreenSafe (GreenSafe Premium, NZYTech).

All the isolates confirmed as *Campylobacter* spp. by the Multiplex PCR, but whose species could not be identified by this method, were submitted to identification by matrix assisted laser desorption ionization-time of flight (MALDI-TOF) mass spectrometry (Vitek^®^ MS, Oxoid). 

### 2.4. Antimicrobial Susceptibility Testing

Antimicrobial susceptibility testing was performed, according to the Kirby–Bauer method. In Mueller–Hinton agar supplemented with 5% defibrinated horse blood (Oxoid) and β-NAD (Sigma, Lisbon, Portugal), a 0.5 McFarland suspension of bacterial culture in saline solution was inoculated and antimicrobial susceptibility testing was performed by disk diffusion, for the following drugs (charge of the disk): ciprofloxacin (5 µg), erythromycin (15 µg), tetracycline (30 µg), gentamicin (10 µg), ampicillin (10 μg) and amoxicillin/clavulanic acid (30 µg) (Oxoid). 

The diameters of the inhibition zones were measured after a 24-h incubation, at 41.5 °C, in a microaerobic environment generated by Anoxomat (Advanced Instruments Inc., Königswinter, Germany). Results were interpretated following the European Committee on Antimicrobial Susceptibility Testing (EUCAST) [17] and the Comité de l’antibiogramme de la Société Française de Microbiologie cut-offs.

### 2.5. Molecular Typing 

Molecular typing was performed using enterobacterial repetitive intergenic consensus PCR (ERIC-PCR), using as template the DNA extracted in the automated nucleic acid extraction platform eMAG™ (bioMérieux, Marcy-l'Étoile, France), and the primers described by Versalovic and colleagues (1991), ERIC-1R (5′-ATGTAAGCTCCTGGGGATTCAC-3′) and ERIC-2 (5′- AAGTAAGTGACTGGGGTGAGCG-3′) [18]. The PCR products were separated in 2% (*w*/*v*) agarose (Agarose Ultrapure grade, NZYtech) in 1× TBE buffer, stained with 1% GreenSafe (GreenSafe Premium, NZYTech).

### 2.6. Whole Genome Sequencing and Bioinformatics Analysis

Whole genome sequencing (WGS) was performed in 10 selected isolates of *C. jejuni*—representative of the variety of ERIC profiles obtained—using the Illumina technology. Each sample was quantified with the Qubit fluorometer (Qubit^TM^ DNA HS Assay, Life Technologies, Thermo Fisher Scientific Inc., Porto Salvo, Portugal). Library preparation was obtained using the Illumina DNA Prep kit (Illumina Inc., San Diego, CA, USA) according to the manufacturer’s manual. The libraries prepared were loaded onto NextSeq 500/550 Mid Output Reagent Cartridge v2, 300 cycles kit (Illumina Inc.) and sequenced on an Illumina NextSeq 500 platform and generated 150 bp paired-end reads.

Genomes were de novo assembled as previously described [19], using the INNUca v4.2.0 pipeline (https://github.com/B-UMMI/INNUca, accessed on 2 February 2021) [20] that consists of several integrated modules for reads QA/QC, de novo assembly and post-assembly optimization steps. The obtained draft genome sequences (of ~1.66 Mbp in length split in no more than 41 contigs for ST-22 strains and of ~1.76 Mbp in length split in no more than 62 contigs for ST-6461 strains) were annotated with the RAST server v2.0 (http://rast.nmpdr.org/, accessed on 8 February 2021) [21]. Raw sequence reads used in the present study were deposited in the European Nucleotide Archive (ENA) under the study accession number PRJEB44937.

In silico Multilocus Sequence Typing (MLST), extended Multilocus Sequence Typing (eMLST) and flaA short variable region (*flaA* svr) typing was determined at the PubMLST platform (https://pubmlst.org/, accessed on 12 February 2021).

In order to understand the phylogenetic positioning of the Portuguese (PT) isolates from dogs, other ST-6461 (*n* = 91) and ST-22 (*n* = 83) (sharing the same *flaA* svr type of the PT dogs), strains from different geographic regions and sources (Appendix A), with available genomes, archived in BIGSdb (accessed on 15 February 2021), were selected, and their paired reads were downloaded from ENA and de novo assembled as described above. In addition, from 48 Portuguese clinical strains collected in 2020, with available genomes, one ST-6461 strain and four ST-22 strains were selected and included in the phylogenetic analysis. The genetic relatedness among strains was evaluated through comparative gene-by-gene analysis using the publicly available *C. jejuni* INNUENDO cgMLST and wgMLST Schemas composed by 678 and 2795 loci, respectively [22]. For all genomes, allele calling was performed against both schemas, using chewBBACA v2.0.16 [23] with default parameters. Exact and inferred matches were used to construct an allelic profile matrix, where other allelic classifications (see https://github.com/B-UMMI/chewBBACA/wiki, accessed on 1 September 2021) were assumed as “missing” loci. Only profiles with no more than 2% of missing loci in either cgMLST were used for subsequent phylogenetic inferences [22]. Minimum spanning trees (MST) were constructed by taking advantage of the goeBURST algorithm [24] implemented in the PHYLOViZ online 2.0 Beta version (http://online2.phyloviz.net, accessed on 25 February 2021) web-based tool [25], based on 100% shared loci between all isolates (i.e., shared-genome MLST). This allows to increase the resolution power for cluster analysis by maximizing the shared genome in a dynamic manner, i.e., for each sub-set of strains under comparison, the maximum number of shared loci between them is automatically used for tree construction. Whenever it was useful, allelic distances were expressed as percentages of allele differences (AD), expressed as the number of allelic differences over the total number of shared loci under comparison.

### 2.7. Data Analysis

Data regarding the variables: setting, age, sex, and display of diarrhea underwent univariate analysis for each canine population, using Fisher’s exact test given the small sample size. *Campylobacter* spp. status (positive or negative) was used as the dependent variable. A *p*-value of <0.05 was considered significant.

## 3. Results

### 3.1. Frequency and Distribution of Campylobacter spp.

Among the 125 rectal swabs collected from 110 dogs, 26 (20.8%; 95% confidence interval (CI) = 14.6–28.7%) had a positive culture for *Campylobacter* spp., and a total of 32 isolates was obtained (Table 1). There was a higher frequency in hunting dogs, with 19/54 (35.2%; 95% CI = 23.8–48.5%) of positive samples, than in companion dogs, with *Campylobacter* spp. being isolated in 7/71 (9.9%; 95% CI = 4.9–19.0%) of the samples collected; this difference was statistically significant (*p* < 0.05). Regarding the distribution of *Campylobacter* species, *C. jejuni* (*n* = 14) was the predominant species, with most isolates (12/14) being obtained from hunting dogs, and the remaining (2/14) from dogs presented for consultation, one at the Veterinary Clinic and the other at the Veterinary Hospital; only the latter showed clinical signs. The second most isolated species was *Campylobacter lari* with 13 isolates obtained from healthy boarhounds. Four *C. upsaliensis* isolates were found, three of them from diarrheic animals presented at the Veterinary Hospital, and one from a non-diarrheic dog sampled at the Veterinary Clinic. Finally, a single *C. coli* isolate was detected in a non-diarrheic dog at the Veterinary Clinic. Mixed colonization was observed in six animals, five of them simultaneously presenting *C. jejuni* and *C. lari* and another one carrying two different strains of *C. jejuni*. The only significant association observed was between the presence of *C. jejuni* and *C. upsaliensis* and the display of diarrhea in companion dogs from the urban setting (*p* < 0.05). Apart from the apparent association with clinical signs, no other surveyed variable showed significant differences (*p* < 0.05).

Substantial differences were observed when comparing the two sampling moments of the boarhounds (Appendix A). Regarding the first sampling event, *C. lari* was the most frequently identified Campylobacter species (7/9) followed by *C. jejuni* (2/9), both obtained from animals that had recently been hunting. In the second sampling moment, performed a month later, after all the dogs had been hunting, there was an increase in the number of isolates obtained (*n* = 14), especially regarding *C. jejuni*, which became the most frequently identified species (8/14). Tendentially, dogs held the same *Campylobacter* spp. status (positive or negative) in both moments. Only in two animals with a positive status in the first sampling time, no isolate was obtained in the second moment, and regarding the dogs with a negative status in the first sample, only one later tested positive. It should be noted that, in the second collection, most of the animals presenting *Campylobacter* had mixed colonization (6/8), a fact not observed in the first moment.

### 3.2. Campylobacter jejuni

Considering that *C. jejuni* was the species more frequently found in this study and being the main responsible for human infection, we focused our study on this species. Of the total *C. jejuni* isolates (*n* = 14), only seven were identified at species level by PCR Multiplex and the identification of the remaining was carried out by MALDI-TOF. All *C. jejuni* isolates not identified by the multiplex PCR belonged to ST-22. Several mismatches were detected in the primer annealing sequence of the target gene (*cj04141*) preventing the species identification of these isolates (data not shown). Antimicrobial susceptibility testing revealed a high proportion of *C. jejuni* isolates resistant to ciprofloxacin (93%), tetracycline (64%), and ampicillin (57%). No resistance to erythromycin, gentamicin, or amoxicillin/clavulanic acid was observed. Altogether, five different drug resistance profiles of *C. jejuni* were found, with three isolates being multidrug resistant (ciprofloxacin, tetracycline, and ampicillin). 

ERIC-PCR fingerprints of the 13 isolates tested showed seven different profiles (Figure 1), which could be grouped into four clusters based on their similarities; the first cluster (C1) included a single isolate from the boarhounds; the second one (C2) comprised three isolates from hunting dogs: one from the pack of boarhounds and two from the same pack of small game hounds. Cluster 3 (C3) consisted of three isolates obtained in the second sample collection of boarhounds, and finally, cluster 4 (C4) comprised five isolates from the boarhounds and one from the Veterinary Hospital. 

To further explore genetic diversity, 10 isolates, representative of the variety of ERIC profiles obtained, were subjected to WGS. The in silico MLST typing showed that the 10 isolates were distributed in five STs: ST-22 (*n* = 4), ST-6461 (*n* = 3), ST-48 (*n* = 1), ST-148 (*n* = 1), and ST-8569 (*n* = 1). The in silico *flaA* short variable region (svr) identified eight different alleles, with three isolates displaying two *flaA* svr alleles. A correlation between *flaA* svr and MLST typing was observed regarding ST-22, as all isolates had the same allele (*flaA*-161). The same was not observed among those belonging to the ST-6461 where a *flaA* svr variety was detected, with the isolates belonging to *flaA*-67, *flA*-49/395, and *flA*- 49.

In general, the clustering of the isolates was highly consistent between different typing methods (Table 2); in fact, except for C2, which hosted two different STs (ST-8569 and ST-148), each ERIC-PCR cluster was correlated with a specific ST (C1: ST-48, C3: ST-6461 and C4: ST-22).

When analyzing the *C. jejuni* isolates from boarhounds, a diversity of *C. jejuni* strains was observed among the different animals from the same pack, and among the same animals in the two sampling events (Appendix A). The two isolates obtained in the first moment belonged to two different STs (ST-48 and ST-8569) that were not identified in the second sampling moment. In this case, isolates were distributed by ST-22 (*n* = 5) and ST-646 (*n* = 3), with one dog harboring *C. jejuni* isolates from two different STs.

### 3.3. Comparison with Human Isolates

The genomes of the dog strains from the two most frequent STs (ST-6461, *n* = 3; ST-22, *n* = 4) were analyzed together with those of strains from humans isolated in Portugal, during 2020 from the same STs (ST-6461, *n* = 1; ST-22, *n* = 4; Appendix A, respectively), through ERIC-PCR, eMLST and wgMLST. Similar genomic relationships were pointed by all methods used, with a high degree of congruency between the results of ERIC-PCR (Figure 2) and eMLST (Appendix A). The analysis of eMLST allowed the detection of differences between isolates from the same ST. Regardless of the source, these differences were also observed in the ERIC profile, as variations in a small number of loci could be associated with minor changes in the ERIC profile (Figure 2). 

All three dog ST-6461 PT isolates exhibited the same wgMLST profile, and were genetically closer to the PT human isolate, composing a genetic cluster with only 14 allelic differences between them (Figure 3A). When compared with the remainder ST-6461 strains, a median of 38 allelic differences is seen (Figure 3B). 

Regarding ST-22 (Figure 4), PT isolates were also found to be genetically closer to each other (<13 AD), composing an independent cluster that distances by at least 24 alleles from the remaining ST-22 isolates. The only exception was one PT human isolate (4093) that, differing by 281 alleles, integrated a completely distinct cluster. Interestingly, two dog isolates (Br4/2 and Br9) shared the same allelic profile as a human isolate (4020). 

## 4. Discussion

All *Campylobacter* species found in this study were previously reported in dogs [6,26]. Although a significantly higher frequency was observed in hunting dogs, these results should be viewed with caution and assumptions should not be made due to the small sample size and the second sampling in boarhounds. Despite the small sample size, in companion dogs from the urban setting, an association between diarrhea and pathogenic species of *Campylobacter* was found as described in the literature [27,28]. 

Regarding the detection of mixed colonization in boarhounds, the fact that the second samples were obtained further along in the hunting season, when the animals had already been exposed to a greater diversity of environments and animals, could possibly justify this observation.

Besides being the only species detected in all groups, *C. jejuni* was the most frequently isolated species in this study. These results are relevant from a public health point of view, since *C. jejuni* is the leading cause of human infections [4].

The fact that none of the *C*. *jejuni* isolates belonging to the ST-22 (6/14) could be identified by the Multiplex PCR is also relevant. This highlights the value of a careful choice of molecular targets that allows a generic detection of isolates and correct species identification. The unsuccessful PCR detection can greatly impact the diagnosis of infection, especially in cases where a direct molecular diagnosis is performed in the biological sample. This may lead to an oversight of certain *Campylobacter* strains, as could be the case with this ST, highly prevalent in our dog isolates.

The predominant *Campylobacter* species isolated in dogs are *C. jejuni* and *C. upsaliensis*, however, there are some divergences between studies on which is the most prevalent [26,27,29]. The fact that the classic diagnostic methods have been optimized aiming at *C. jejuni* and *C. coli* means that they are often not suitable for the isolation of more fastidious species, which are frequently susceptible to the antibiotics present in the selective media and may need different temperatures or times of incubation [30,31]. In this study, in addition to using an incubation temperature of 41.5 °C, higher than the optimal temperature for *C. upsaliensis* (37 °C), the high concentration of Cefoperazone (20 mg/L) in the enrichment broth (above the previously described MIC for the majority of this species [32]), could explain why the number of isolates obtained was lower than the usually reported [6,29,33,34].

In dogs, *C. coli* is reportedly less frequent compared to *C. jejuni* [6,35] and *C. upsaliensis* [6], this fact is in line with the results from this study, where only one isolate was obtained. Regarding *C. lari*, despite the high number of isolates found in this study, they were all obtained from the same group of dogs (boarhounds), the majority from the same animals sampled twice. This can thus form a bias that may explain the unusually high prevalence, as in dogs the prevalence reported is usually very low (1–2%) [6].

Regarding antimicrobial resistance, previous studies in dogs have reported resistance to ciprofloxacin (9–58%), tetracycline (12–32%), and erythromycin to a lesser extent (0–12%) [7,36,37,38]. Although no resistance to erythromycin was observed in this study, higher resistance rates to ciprofloxacin and tetracycline were found (93% and 64%, respectively), and the results are very similar to those recently reported by EFSA regarding *C. jejuni* from humans in Portugal, where rates of 93% resistance to ciprofloxacin and 78% to tetracycline were described [4]. This highlights the dissemination of antimicrobial resistance between different hosts, reinforcing the need for a comprehensive approach to this problem.

When analyzing the isolates from boarhounds, as it was possible to confirm with phenotypic (antimicrobial susceptibility) and genotypic (ERIC-PCR and WGS) assessment, there was a high diversity of *C. jejuni* strains between animals from the same pack and between the same animals in the two sampling events (one month apart). Along with the fact that the bacterium was only isolated from animals that had recently been hunting, this could possibly mean that there is a higher risk of transmission through the environment (while hunting) than between dogs or through feeding, inside the kennel. For this to be confirmed, further studies should be done, not only including more sampling events over time but also covering other boarhound packs. 

Among the five different STs identified in the dog isolates, four of them (ST-22; ST-48; ST-148 and ST-6461) represented STs widely found in humans, suggesting a genetic proximity between the isolates present in both species. This relationship becomes even more relevant through wgMLST, where the PT isolates from ST-22 and ST-6461 formed high proximity clusters with minimal allelic differences. 

The genetic proximity observed between samples of the two different hosts could not be associated with any apparent epidemiologic relationship, as the same wgMLST allelic profile was obtained in a human sample collected in July 2020, 380 km away from the boarhound sample collected in November 2020. The same was observed regarding different canine populations, as the same wgMLST allelic profile was obtained from a sample of a companion dog in an urban area, collected in September 2020, and from a sample of a boarhound living more than 100 km, in November 2020. 

Four different scenarios can justify the presence of identical strains in humans and dogs, as previously reviewed by Mughini-Gras and colleagues (2013). They can both become infected from the same source or from different sources carrying the same strain, humans can become infected from dogs, or dogs can become infected by humans. Unlike the study of Mughini-Gras, where there was a known close proximity between both hosts (comparison between dogs and their owners), in the present study, there was no apparent relationship between the human and dog samples. As so, both hosts becoming infected by different sources carrying the same strain would best seem to explain these findings, even considering the genetic diversity of *Campylobacter* and the geographic distribution of the samples. Thus, this study raises questions regarding the distribution of *Campylobacter* strains and the transmissibility between the two hosts.

Besides being the most common among dogs, in this study, ST-22 presented the highest genetic similarity with the human isolates. In addition, this particular ST has been highly associated with chronic complications such as the Guillain–Barré syndrome, being overrepresented amongst patients with this post-infection complication [39,40,41]. Considering that this ST is rarely found in poultry, the most common animal source of *C. jejuni*, [39,42,43], dogs could be a more relevant reservoir than currently considered. 

Improved detection and control of clinically relevant *Campylobacter* species will be achieved through an integrated One Health approach that includes an appropriate selection of typing techniques. ERIC-PCR, which in this study revealed a strong correlation with MLST, seems to be a very suitable option for routine genotyping analysis, although it still needs optimization and validation with a greater number of isolates. Given its effortlessness and low cost, this technique may be of special interest regarding sources of *Campylobacter* not normally studied, as it is the case with dogs.

In conclusion, despite being the first insight into *Campylobacter* spp. in dogs in Portugal, this study revealed surprising genetic proximity (evident through wgMLST) between the *C. jejuni* isolates obtained in dogs and those isolated from geographically distant humans, without any apparent epidemiological relationship. Thus, regardless of the exact transmission route, which is of greater interest to determine, the overlap of *Campylobacter* strains between two hosts with ever closer contact in today’s society, requires an in-depth study to assess the real impact that this relationship may represent in campylobacteriosis.

## Figures and Tables

**Figure 1 microorganisms-09-02231-f001:**
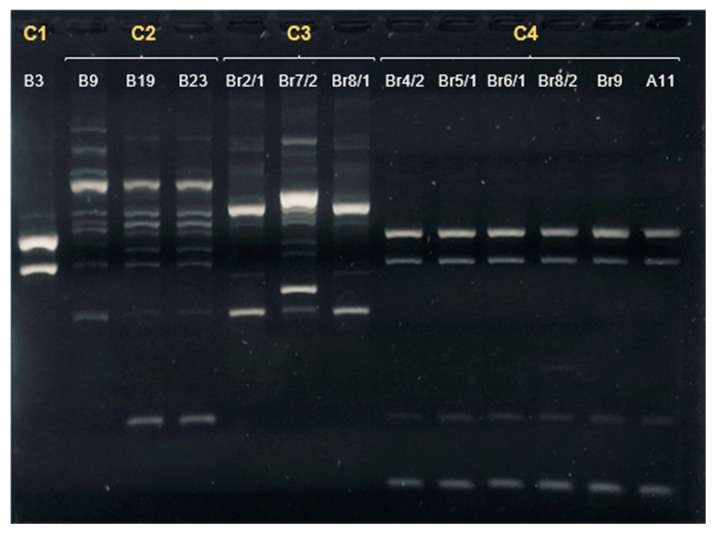
ERIC-PCR fingerprints of *Campylobacter jejuni* isolated in dogs. The ID of the sample is presented in white. The letter and numbers in yellow represent the cluster assigned to the profile.

**Figure 2 microorganisms-09-02231-f002:**
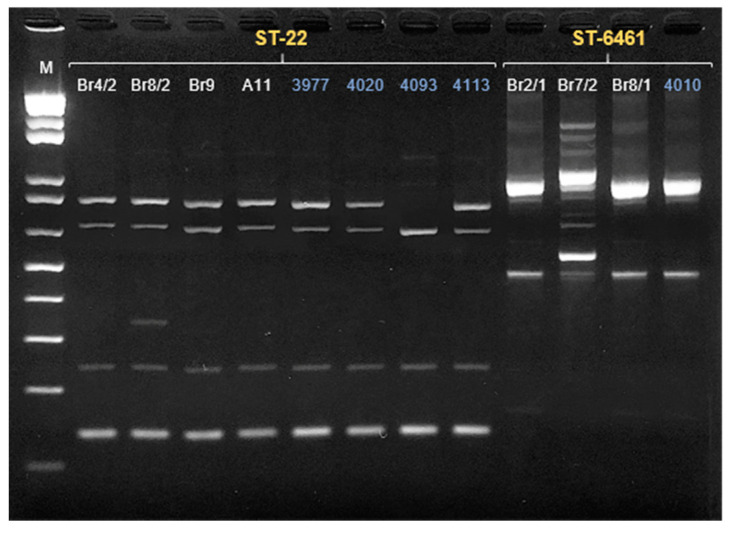
ERIC-PCR fingerprints of *Campylobacter jejuni* ST-22 and ST-6461 (presented in yellow) in a 2% agarose gel. Lane M, 1 Kbp Plus ladder (Invitrogen, Waltham, MA, USA). The ID of the sample is presented in white in the case of the dog isolates and in blue in the human isolates. Finally, a comparative gene-by-gene analysis was performed between PT isolates and other ST-6461 (*n* = 91) and ST-22 (*n* = 83) with available genomes from different regions and sources (Appendix A), with the results represented by minimum spanning trees (MSTs) (Figure 3 and Figure 4).

**Figure 3 microorganisms-09-02231-f003:**
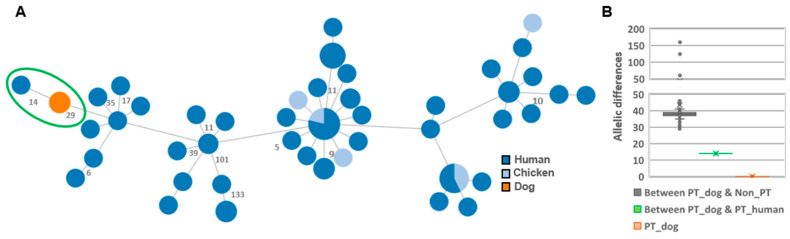
Genomic relatedness among *Campylobacter jejuni* ST-6461 strains based on a dynamic gene-by-gene approach using a wgMLST schema with 2795 loci. (**A**) The Minimum spanning tree was constructed using the goeBURST algorithm implemented in the PHYLOViZ online platform and is based on the allelic diversity found among the 1011 genes shared by 100% of the validated strains. Filled circles (nodes) represent unique allelic profiles, and are colored according to strains’ source (human, chicken, or dog). The size of nodes is proportional to the number of isolates it represents. Portuguese strains are marked with a green circle: three isolates from dogs (PT_Cj-Br2/1, PT_Cj-Br7/2, PT_Cj-Br8/1) and one from human (PT_Cj-4010). The numbers in grey on the connecting lines represent the allele differences (AD) between strains. For better visualization, only AD ≥ 5 are shown. (**B**) Box plot depicting the AD (including the median values) within and between PT strains from dogs and the remainder.

**Figure 4 microorganisms-09-02231-f004:**
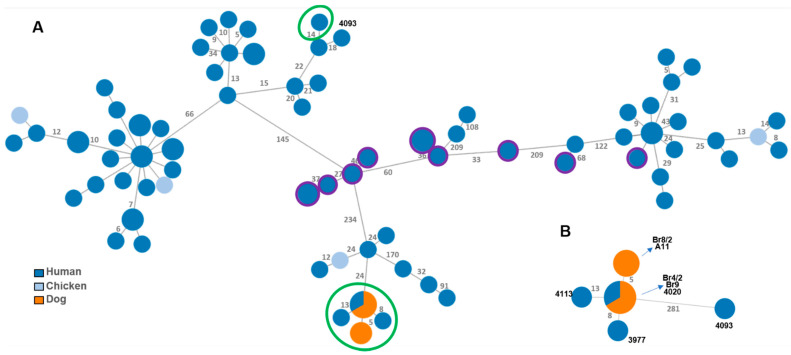
Phylogeny of *Campylobacter jejuni* ST-22 strains based on a dynamic gene-by-gene approach using a wgMLST schema with 2477 loci. (**A**) The initial MST was constructed based on the allelic diversity found among the 924 genes shared by 100% of the validated strains. Portuguese strains are marked with green circles: four from dogs (A11, Br8/2, Br4/2, Br9) and four from humans (3977, 4020, 4093, 4113); US strains are marked with purple outer rings; the remaining strains are from the UK. (**B**) Sub-MST reconstruction based on the allelic diversity found among the 953 genes shared by all validated PT strains (ID strains in bold). For both panels, trees were constructed using the goeBURST algorithm implemented in the PHYLOViZ Online platform. Filled circles (nodes) representing unique allelic profiles are colored according to strains’ source (human, chicken or dog). The size of nodes is proportional to the number of isolates it represents. The numbers in grey on the connecting lines represent the allele differences (AD) between strains. For better visualization, only AD ≥ 5 are shown.

**Table 1 microorganisms-09-02231-t001:** Characteristics of the samples collected (*n* = 125) and species distribution among *Campylobacter* isolates (*n* = 32), according to canine populations.

N-Total [n1-Companion/n2-Hunting]	Gender		Age		Diarrhea		Setting	
	Male	Female	0–2	>2	Yes	No	Urban	Rural
N. dogs	55	55	42	68	12	98	50	60
	[32/23]	[39/16]	[23/19]	[44/24]	[12/0]	[59/39]	[50/0]	[21/39]
N. samples	63	62	47	78	12	113	50	75
	[32/31]	[39/23]	[23/24]	[44/34]	[12/0]	[59/54]	[50/0]	[21/54]
Distribution by species								
*C. jejuni*	10	4	4	10	1	13	1	13
	[2/8]	[0/4]	[2/2]	[0/10]	[1/0]	[1/12]	[1/0]	[1/12]
*C. upsaliensis*	2	2	2	2	3	1	3	1
	[2/0]	[2/0]	[2/0]	[2/0]	[3/0]	[1/0]	[3/0]	[1/0]
*C. lari*	6	7	5	8	0	13	0	13
	[0/6]	[0/7]	[0/5]	[0/8]	[0/0]	[0/13]	[0/0]	[0/13]
*C. coli*	1	0	1	0	0	1	0	1
	[1/0]	[0/0]	[1/0]	[0/0]	[0/0]	[1/0]	[0/0]	[1/0]
Total	19	13	12	20	4	28	4	28
	[5/14]	[2/11]	[5/7]	[2/18]	[4/0]	[3/25]	[4/0]	[3/25]

**Table 2 microorganisms-09-02231-t002:** ERIC-PCR, Sequence type, Clonal complex and *flaA svr* of *Campylobacter jejuni* isolates from dogs studied by whole genome sequence.

Sample	Date of Isolation	Group	ERIC-PCR Cluster	MLST Sequence Type	Clonal Complex	*flaA svr*
A11	24/09/2020	Companion/Urban	4	22	22	161
B3	23/10/2020	Hunting/Rural	1	48	48	32;102
B9	23/10/2020	Hunting/Rural	2	8569	179	571
B19	6/11/2020	Hunting/Rural	2	148	21	36;36
Br2/1	20/11/2020	Hunting/Rural	3	6461	353	67
Br4/2	20/11/2020	Hunting/Rural	4	22	22	161
Br7/2	20/11/2020	Hunting/Rural	3	6461	353	49;395
Br8/1	20/11/2020	Hunting/Rural	3	6461	353	49
Br8/2	20/11/2020	Hunting/Rural	4	22	22	161
Br9	20/11/2020	Hunting/Rural	4	22	22	161

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
