# Peer review of "Campylobacter jejuni in Different Canine Populations: Characteristics and Zoonotic Potential"

_microorganisms, 2021, doi:10.3390/microorganisms9112231_

Round 1

Reviewer 1 Report

The manuscript describes first data on prevalence of Campylobacter spp. in dogs in Portugal. Several different dog populations were sampled and basic epidemiological associations with presence in faeces were evaluated. The isolates were characterised by both phenotypic and more comprehensively by genotypic methods. More detailed study was conducted for C. jejuni isolates for which the authors added isolates from humans in Portugal and isolates from humans and chicken from other countries available in the PubMLST database. 

Materials and methods

Please make a section in materials and methods for statistical and exploratory analysis with description of tools and methods used. This is very important for transparency and reproducibility.

Line 124 - add EUCAST version/year by which the results were interpreted.

Results

Since in the section of frequency and distribution of Campylobacter spp. there are two main aspects of comparisons, one of hunting dogs vs. companion dogs and the other one of dogs with and without  diarrhoea, the authors could reverse the format of the Table 1 and separate data from companion and hunting dogs and then show within each population the results by gender, age, diarrhoeal signs presence and setting of animals. So, a reversal of rows and columns. Then within each cell one can write results, for instance, for companion population and for hunting one in brackets or some other division like | or \. No point in making columns for both, the readability of the table would be affected. This would also help a lot with clarity of which dogs (sub)populations had which species. Currently it is hard to discern the structure of that data between what is written in the text and  presented in the table. Even more important is that the numbers in the table do not add up, for instance C. upsaliensis data for group variable, Campylobacter spp. data for age.

Why are not results from two sampling times of hunting dogs presented?

Please highlight the differences between eMLST and ERIC-PCR typing in Supplementary table 1. One can highlight with bold, italics or underline the rows where differences exist to make it easier for the reader. For instance, ST-22 data for filvD and fmutY...

Overall the authors really nicely showed the congruence of typing methods for their subset of C. jejuni isolates.

With regard to the molecular data in the section of comparison with human isolates, why is there no cgMLST results presented that are described in the methods? Why only wgMLST? 

In line 249, is this 38 or should it be 29 allelic differences? But isn't it odd that all 3 dog ST-6461 isolates had exact wgMLST profile yet 3 different profiles are seen with flaA typing scheme? This should be discussed.

Typo in line 251, the authors I presume mean 24 allelic differences.

Discussion

There are paragraphs that belong in results section and not in discussion. The unsuccessful PCR detection of ST-22 isolates that were confirmed by MALDI is of interest and should not be first mentioned in the discussion. Have other studies observed similar issues? This is really interesting from diagnostic point of view regarding isolate taxonomic identifications. The other instance of discussion of results not presented is about the data from two sampling times in lines 313-321. Report similarities between packs and between times of isolation in results if this is to be discussed. 

The association of diarrhoea and Campylobacter spp. is very simplistic. What is the basis of grouping all species of Campylobacter genus as pathogens. There are many studies that showed not all species of the genus should be treated as pathogenic. The authors only pointed to the limitation of the sample size and of the double sampling of 15 dogs.

Why is there no more detailed discussion of unusually low prevalence of C. upsaliensis? Expand what you mention about time and temperature of incubation conditions? Is 41,5°C favourable for C. jejuni and C. lari but not C. upsaliensis? Given the enrichment step and then another 2 days of cultivation, most studies having successful isolation of C. upsaliensis also had similar time of incubation? Is it a problem of Bolton broth and/or antimicrobials in mCCDA? Please see studies by Aspinal et al, Acke, Bojanic...

In my opinion the authors are overconjecturing the conclusions from observation of genetic similarity between dog and human isolates for which the data and analyses are questionable and problematic for several reasons. Inspection of PubMLST database for ST-22 and ST-6461 shows bias in selection of isolates. That is, the selection of ST-6461 seems OK as these are only available from Europe, most of which from humans and then to a lesser extent to chicken and only 1 isolate from a lamb (106 isolates in total). On the other hand, the database has over 1,100 isolates of ST-22, spanning all continents and numerous sources like humans, chicken, cattle, sheep, pig, ducks, other animals and various environmental isolates. The authors, apart from their dog and human isolates from Portugal, used only 4 chicken isolates from UK, 11 human isolates from USA and the remainder from humans from UK. This bias may easily affected results so the Portugal isolates look more similar to each other than it could be if a more representative sample of source diversity were used.

Out of 4 scenarios of transmission routes between sources and hosts the authors discuss, the scenario for their results of both hosts becoming infected by different sources carrying the same strain or even same source is most likely. Here the source is not meant in light of point-source, the actual infection of a human whose data is in the database by the actual animal which was the source of isolate within the PubMLST database. No one should think that the person who got ill in July 2020 has actually had contact with that particular dog who was positive in November 2020. Sources are here considered on a population level. I can see, and praise, the addition of context of amplifying hosts in molecular epidemiology but the authors should consider that in their inference about dog-human transmission route. The question is for hosts like dogs and humans who are not primary reservoirs of Campylobacter in nature (the amplifying hosts), where did they get it from, as in which animal is likely to have these genotypes? Some are generalists and some are specialists with regard to animal species we observe them. Clearly the dogs and humans can contract it from each other, and if the authors had a within household study like Lapo did, it would be very much more supportive of that transmission route. The authors, on the other hand, used heterogenous data from PubMLST and they cannot claim in comparisons using various countries and animals then that their Portugal isolates do not have any epidemiologic relationship. When comparing from such sample, being from the same country is an epidemiological relationship. All isolates of any ST from one country could be expected to be more similar to each other than to isolates of that same ST from somewhere else on the globe. Especially, if comparison is on a whole-genome level. In this regard, if the authors had taken a larger and more representative diversity of ST-22 and then compared the results of whole-genome versus core-genome MLST, this would be more likely expected. Accessory genome of a population of C. jejuni in one country might be different in another region of the world and thus affect these results and core genome MLST using over 4x less loci but more importantly, being the core loci of all members could help in elucidating these relationships. 

More importantly, the authors wrongly state that chicken are rare sources of ST-22 and that dogs could be more important than poultry. This is an overstatement as I have checked the numbers in the database and currently out of 1,105 of ST-22 isolates, only 0.9% are from dogs whereas both cattle and chicken each account for 5-6% of isolate entries, if anything that is 5x more in both animal sources than in dogs. Then another 1.5% is from other animals, that is interesting. And ~40% of isolates do not have  source stated. This is interesting as the majority is unknown. One can wonder if there are animal reservoirs we haven't identified yet from which maybe both humans and dogs most commonly contract these STs? Perhaps certain wild bird species harbour these and both dogs and people contract it through environmental exposure and not maybe direct contact with that animal species? The database shows that slightly more, ~1% of ST-22 isolates come from environmental waters, farm slurry or farm environment. Perhaps ST-22 is not a specialist ST that does not have a yet unidentified primary source attribution. Maybe it is a generalist like many other STs that one can observe in many different species and likelihood of an end-hosts to become infected with such genotype depends on the varying probabilities it has for exposures to each of those sources.

Lastly, about this issue in molecular work, the sample size is really small. Portugal isolates are comprised of 3 ST-6461 and 4 ST-22 and the rest of samples taken from the public database that as mentioned above, seems to have a selection bias.

Line 353-355 reference is needed for the statement about ERIC-PCR and MLST.

Author Response

Microorganims-1409975

We really appreciated the critical reading and careful review of our manuscript, and greatly acknowledge the comments provided by the reviewers, which were very helpful in improving the quality of the manuscript.

Then, we answer point by point to each one of the remarks raised by the reviewers.

REVIEWER 1:

General comments: The manuscript describes first data on prevalence of Campylobacter spp. in dogs in Portugal. Several different dog populations were sampled and basic epidemiological associations with presence in faeces were evaluated. The isolates were characterised by both phenotypic and more comprehensively by genotypic methods. More detailed study was conducted for C. jejuni isolates for which the authors added isolates from humans in Portugal and isolates from humans and chicken from other countries available in the PubMLST database

Materials and methods

Comment: Please make a section in materials and methods for statistical and exploratory analysis with description of tools and methods used. This is very important for transparency and reproducibility.

Response: This was an oversight we would like to apologise for. This information has been added to the materials and methods, section 2.7.

Comment:
Line 124 - add EUCAST version/year by which the results were interpreted.

Response: This information has been added.

Results

Comment: Since in the section of frequency and distribution of Campylobacter spp. there are two main aspects of comparisons, one of hunting dogs vs. companion dogs and the other one of dogs with and without diarrhoea, the authors could reverse the format of the Table 1 and separate data from companion and hunting dogs and then show within each population the results by gender, age, diarrhoeal signs presence and setting of animals. So, a reversal of rows and columns. Then within each cell one can write results, for instance, for companion population and for hunting one in brackets or some other division like | or \. No point in making columns for both, the readability of the table would be affected. This would also help a lot with clarity of which dogs (sub)populations had which species. Currently it is hard to discern the structure of that data between what is written in the text and presented in the table. Even more important is that the numbers in the table do not add up, for instance C. upsaliensis data for group variable, Campylobacter spp. data for age.

Response: We are very grateful for the suggestion and we altered the structure of Table 1 in accordance. We believe is now easier to interpret the results for each dog (sub)population. Since it didn’t show any significant differences in statistical analysis and to simplify the readability of the table, dogs are now categorized as < 2 years old or ≥ 2 years. We corrected the typos that did not add up.

Comment:
Why are not results from two sampling times of hunting dogs presented?

Response: As we consider this information to be of value, we added this data to the results and presented a new supporting table (Supplementary Table 1) with scrutinized results.

Comment:
Please highlight the differences between eMLST and ERIC-PCR typing in Supplementary table 1. One can highlight with bold, italics or underline the rows where differences exist to make it easier for the reader. For instance, ST-22 data for filvD and fmutY…

Response: We appreciate the suggestion as it facilitates interpretation and we have highlighted these differences.

Comment:
Overall the authors really nicely showed the congruence of typing methods for their subset of C. jejuni isolates.

Response: We really appreciate this comment.

Comment:
With regard to the molecular data in the section of comparison with human isolates, why is there no cgMLST results presented that are described in the methods? Why only wgMLST?

Response:  cgMLST was only used to select strains to be used in wgMLST. Indeed, according to INNUENDO pipeline (please see ref. 19), only profiles with no more than 2% of missing loci in the cgMLST were used for subsequent phylogenetic inferences, as it is stated in the manuscript.

Comment: In line 249, is this 38 or should it be 29 allelic differences? But isn't it odd that all 3 dog ST-6461 isolates had exact wgMLST profile yet 3 different profiles are seen with flaA typing scheme? This should be discussed.

Response: The number described, 38, refers to the median allelic differences, as derived from Figure 3B. To be clearer, we now mention Figure 3B next to the number of allelic differences. Regarding flaA, it is not included neither in the cgMLST nor in the wgMLST INNUENDO schemas for Campylobacter jejuni.

Comment: Typo in line 251, the authors I presume mean 24 allelic differences.

Response: Thank you for pointing out this error. The correction was made.

Discussion

Comment: There are paragraphs that belong in results section and not in discussion. The unsuccessful PCR detection of ST-22 isolates that were confirmed by MALDI is of interest and should not be first mentioned in the discussion. Have other studies observed similar issues? This is really interesting from diagnostic point of view regarding isolate taxonomic identifications. The other instance of discussion of results not presented is about the data from two sampling times in lines 313-321. Report similarities between packs and between times of isolation in results if this is to be discussed.

Response: We are grateful to the reviewer for these suggestions, and we proceeded in accordance. Regarding the unsuccessful detection of ST-22, we also believe that this is really interesting and relevant, especially for laboratories that rely on molecular diagnosis only. We did not find similar issues previously described, probably because it is not very common to do recheck species identification with another method like we did with MALDI-TOF.  Regarding the data from the two sampling times of the boarhounds, as previously mentioned, we consider this information to be of value and added the data to the results.

Comment: The association of diarrhoea and Campylobacter spp. is very simplistic. What is the basis of grouping all species of Campylobacter genus as pathogens. There are many studies that showed not all species of the genus should be treated as pathogenic. The authors only pointed to the limitation of the sample size and of the double sampling of 15 dogs.

Response: We agreed that the details of the association with diarrhea were not very clear. As so, we described this association more explicitly in the results but did not go into great detail in the discussion, given the limitations already described.

Comment: Why is there no more detailed discussion of unusually low prevalence of C. upsaliensis? Expand what you mention about time and temperature of incubation conditions? Is 41,5°C favourable for C. jejuni and C. lari but not C. upsaliensis? Given the enrichment step and then another 2 days of cultivation, most studies having successful isolation of C. upsaliensis also had similar time of incubation? Is it a problem of Bolton broth and/or antimicrobials in mCCDA? Please see studies by Aspinal et al, Acke, Bojanic...

Response: In accordance with the comment, and consulting the articles kindly suggested by the reviewer, we extended the discussion regarding the low prevalence of C. upsaliensis.

Comment: In my opinion the authors are overconjecturing the conclusions from observation of genetic similarity between dog and human isolates for which the data and analyses are questionable and problematic for several reasons. Inspection of PubMLST database for ST-22 and ST-6461 shows bias in selection of isolates. That is, the selection of ST-6461 seems OK as these are only available from Europe, most of which from humans and then to a lesser extent to chicken and only 1 isolate from a lamb (106 isolates in total). On the other hand, the database has over 1,100 isolates of ST-22, spanning all continents and numerous sources like humans, chicken, cattle, sheep, pig, ducks, other animals and various environmental isolates. The authors, apart from their dog and human isolates from Portugal, used only 4 chicken isolates from UK, 11 human isolates from USA and the remainder from humans from UK. This bias may easily affected results so the Portugal isolates look more similar to each other than it could be if a more representative sample of source diversity were used.

Response: In order to understand the phylogenetic positioning of the Portuguese isolates from dogs, we included strains from different geographic regions and sources. Accessing PubMLST on 15th of February, the BIGSdb was downloaded and strains from ST-6461 or ST-22 were selected based on availability of paired reads in ENA. In addition, for ST-22, we only selected strains with the same flaA genotype. Therefore, in total we included 82 strains from ST-22 and 91 from ST-6461. To be clearer, we added this information to the materials and methods, section 2.6.

Comment: Out of 4 scenarios of transmission routes between sources and hosts the authors discuss, the scenario for their results of both hosts becoming infected by different sources carrying the same strain or even same source is most likely. Here the source is not meant in light of point-source, the actual infection of a human whose data is in the database by the actual animal which was the source of isolate within the PubMLST database. No one should think that the person who got ill in July 2020 has actually had contact with that particular dog who was positive in November 2020. Sources are here considered on a population level. I can see, and praise, the addition of context of amplifying hosts in molecular epidemiology but the authors should consider that in their inference about dog-human transmission route. The question is for hosts like dogs and humans who are not primary reservoirs of Campylobacter in nature (the amplifying hosts), where did they get it from, as in which animal is likely to have these genotypes? Some are generalists and some are specialists with regard to animal species we observe them. Clearly the dogs and humans can contract it from each other, and if the authors had a within household study like Lapo did, it would be very much more supportive of that transmission route.

Response: In fact, we share the same perspective as the reviewer. However, during the discussion we felt that we should not try to enumerate the potential amplification and transmission pathways that might be involved; firstly, because there are many (already known); secondly because there are still many to be known; and lastly because, as pointed out by the reviewer, we lack research concerning Campylobacter transmission either in the context of cohabitation or via environmental sources. Therefore, we rather prefer to mention sensu lato the four possible transmission pathways and underline the higher probability that Campylobacter jejuni may circulate in the environment, and that further extensive research is needed to identify the animal species where Campylobacter may be amplified and the points in the environment where it may persist.

Comment: The authors, on the other hand, used heterogenous data from PubMLST and they cannot claim in comparisons using various countries and animals then that their Portugal isolates do not have any epidemiologic relationship. When comparing from such sample, being from the same country is an epidemiological relationship. All isolates of any ST from one country could be expected to be more similar to each other than to isolates of that same ST from somewhere else on the globe. Especially, if comparison is on a whole-genome level. In this regard, if the authors had taken a larger and more representative diversity of ST-22 and then compared the results of whole-genome versus core-genome MLST, this would be more likely expected. Accessory genome of a population of C. jejuni in one country might be different in another region of the world and thus affect these results and core genome MLST using over 4x less loci but more importantly, being the core loci of all members could help in elucidating these relationships.

Response: The reviewer raised an important question, for which we had previously checked but it was not mentioned in the first version of the manuscript. Indeed, depending on the ST or CC, lineage-specific accessory genome sharing may be higher among isolates from the same country. On the other hand, the approach we used to infer the phylogenetic relationships among our isolates, although wgMLST based, is a dynamic approach, since it allows maximizing the shared genome in a dynamic manner, i.e., for each sub-set of strains under comparison, the maximum number of shared loci between them is automatically used for tree construction, and at the same time allows to discard any bias related with accessory genome. In order to be clearer, we added this explanation to the materials and methods, section 2.6.

This dynamic approach resulted in a number of loci used much lower than the initial 2795 loci described in the C. jejuni INNUENDO wgMLST Schema: 1011 and 953 loci for the ST-6461 and ST-22 Minimum spanning trees (MST), respectively, as it is described in Figure’s 3 and 4 caption.

In addition, the wgMLST MSTs for both STs under study exactly matches the cgMLST MSTs (please see attached file-cgMLST MST), and either considering the wgMLST MST or the cgMLST MST, a cluster by country is not evident, even for the PT isolates which are in small number (Figure 4A). In order to highlight this message, we reinforced the origin by country in the ST-22 MST (Figure 4A).

Comment: More importantly, the authors wrongly state that chicken are rare sources of ST-22 and that dogs could be more important than poultry. This is an overstatement as I have checked the numbers in the database and currently out of 1,105 of ST-22 isolates, only 0.9% are from dogs whereas both cattle and chicken each account for 5-6% of isolate entries, if anything that is 5x more in both animal sources than in dogs. Then another 1.5% is from other animals, that is interesting. And ~40% of isolates do not have source stated. This is interesting as the majority is unknown. One can wonder if there are animal reservoirs we haven't identified yet from which maybe both humans and dogs most commonly contract these STs? Perhaps certain wild bird species harbour these and both dogs and people contract it through environmental exposure and not maybe direct contact with that animal species? The database shows that slightly more, ~1% of ST-22 isolates come from environmental waters, farm slurry or farm environment. Perhaps ST-22 is not a specialist ST that does not have a yet unidentified primary source attribution. Maybe it is a generalist like many other STs that one can observe in many different species and likelihood of an end-hosts to become infected with such genotype depends on the varying probabilities it has for exposures to each of those sources.

Response: The depth of review of the entire manuscript, as well as the attention to detail that this comment so well demonstrates, has not been indifferent to the authors, who are deeply grateful.

Although in absolute numbers there are 5x more ST-22 isolates from chickens and cows than from dogs, if we look at the total number of isolate entries from these animals in the database, we see that 17.2% are from chickens, 4.1% from cows and only 0.3% from dogs. Comparing the total frequency of isolate entries with the ST-22 isolates in particular, we find that the percentage of ST-22 isolates from dogs (0.9%) is 3x greater than the overall percentage of isolate entries from this source (0.3%). On the other hand, if we look at chickens, which represent 6% of ST-22 isolates, this value is almost 3x lower than the total percentage of chicken isolates (17.2%). In the case of cattle, the percentage of ST-22 is slightly higher than the general one (ST-22=4.5%; overall=4.1%), which although interesting, is not as significant as the difference observed in dogs.

Also interesting, if we consider the different STs present in the database, 1.4% of all isolates belong to ST-22, with a similar percentage observed in human isolates. Regarding chickens, this ST was only identified in 0.5% of the isolates, whereas in dogs, 3.7% of isolates entries belong to ST-22.

Looking at the different ST-22 sources presented in the database, only dogs, cats, “other animals”, “farm slurry + farm environment”, and isolates without stated source showed a percentage substantially higher than expected. For example, water, which is the source of 1.6% of the total samples in the database, only accounts for 0.7% of the ST-22 isolates, less than half of what might be expected. There is indeed still a lot to understand regarding ST-22. As stated by the reviewer, the majority of isolates does not have a known source (1.5% from other animals, and ~40% of isolates without source stated) and the possibility of indirect contraction of Campylobacter from wild animals through environmental exposure cannot be ignored. Even so, ST-22 does seem to be a specialist ST in which dogs (and probably cats too although the total number of isolates in the database belonging to this source is very limited (n=32)) may play a more relevant part than currently considered.

Despite being just speculation (that goes beyond the scope of this study), doing a general analysis of the different CCs present in the database, there seems to be a strong tendency for the most representative CCs in chicken isolates to correspond to the less representative in the dogs, and vice versa, revealing a differentiating trend of the strains circulating in each source.

Thus, despite the small sample size of this work, the genetic relationship observed validates the idea that dogs may play a more important part in the epidemiology of certain STs, as it seems to be the case with ST- 22, the ST to which almost half of the dog samples of this study belonged, and the only one obtained in two different canine populations (hunting dogs from rural setting and companion dog from urban setting).

Comment: Lastly, about this issue in molecular work, the sample size is really small. Portugal isolates are comprised of 3 ST-6461 and 4 ST-22 and the rest of samples taken from the public database that as mentioned above, seems to have a selection bias.

Response: Indeed, the sample size from Portugal is small since the initial number of isolates, either from the human side or from the dogs side is limited. Regarding dogs, the total number of isolates is 14; from the human side, we only had 48 isolates, collected during 2020, with available genomes, from which one was from ST-6461 and four from ST-22. In order to be clearer and contextualize the dimension of the Portuguese sample size, we added this information to the materials and methods, section 2.6.

Comment: Line 353-355 reference is needed for the statement about ERIC-PCR and MLST

Response: With this statement we refer to the conclusions observed in our study. We rephrased this part hoping it became more perceptive.

Reviewer 2 Report

Dear Authors,
your study is very well structured both for the very accurate experimental design and for the elaborate discussion of the data that emerged.
Particularly interesting I found:
- the analysis carried out on the isolated strains and the comparison with human strains, underlining the possibility of transmission not only from animals to humans but also vice versa, in line with the One Health approach;
- the risk of transmission linked to contact with animals or humans;
- the possibility of dog infection considering the contexts it frequents;
With regard to the scientific literature consulted and the relative comparative data, the literature concerning the presence of Campylobacter in dogs has recently been enriched with new scientific contributions.
For example, Santaniello and colleagues (2021, International Journal of Environmental Research and Public Health), carried out a study concerning the presence of C. jejuni and also C. coli in dogs to be involved in animal-assisted therapies. and Verma et al, 2021 (BMC Veterinary Research) performed a study on AMR of C. jejuni in shelther dogs. 

In my opinion, with some small corrections but necessary corrections your paper can be published, also attracting great interest from the readers.

Line 32: I suggest removing the name of the species "jejuni", already present in the title, with the abbreviation "spp." to increase the availability of the article.

Lines 195-196: as reported by the WHO, Campylobacter jejuni together with C. coli, are the main culprits of infections in humans. Please supplement this information and support it with a bibliographic reference.

Kind Regards.

Author Response

Microorganims-1409975

We really appreciated the critical reading and careful review of our manuscript, and greatly acknowledge the comments provided by the reviewers, which were very helpful in improving the quality of the manuscript.

Then, we answer point by point to each one of the remarks raised by the reviewer.

REVIEWER 2:

General comments: Your study is very well structured both for the very accurate experimental design and for the elaborate discussion of the data that emerged.

Particularly interesting I found:

- the analysis carried out on the isolated strains and the comparison with human strains, underlining the possibility of transmission not only from animals to humans but also vice versa, in line with the One Health approach;

- the risk of transmission linked to contact with animals or humans;

- the possibility of dog infection considering the contexts it frequents;

Comment: With regard to the scientific literature consulted and the relative comparative data, the literature concerning the presence of Campylobacter in dogs has recently been enriched with new scientific contributions.

For example, Santaniello and colleagues (2021, International Journal of Environmental Research and Public Health), carried out a study concerning the presence of C. jejuni and also C. coli in dogs to be involved in animal-assisted therapies. and Verma et al, 2021 (BMC Veterinary Research) performed a study on AMR of C. jejuni in shelther dogs.

Response:  Thank you for your suggestion. Accordingly, we updated information and added the respective reference to the manuscript.

Comment: In my opinion, with some small corrections but necessary corrections your paper can be published, also attracting great interest from the readers.

Comment: Line 32: I suggest removing the name of the species "jejuni", already present in the title, with the abbreviation "spp." to increase the availability of the article.

Response: We really appreciate the suggestion and we have made the alteration.

Comment: Lines 195-196: as reported by the WHO, Campylobacter jejuni together with C. coli, are the main culprits of infections in humans. Please supplement this information and support it with a bibliographic reference.

Response: We appreciate and agree with the observation. However, given the disproportion between the percentage of infections caused by each of the species (Campylobacter jejuni=83.1%; Campylobacter coli=10.8%), and considering that in this part we intend to emphasize the role of C. jejuni, we supplement the information on the main culprits of infections in humans in the introduction and kept only the reference to C. jejuni in this section.